# Global Buckling Investigation of the Flanged Cruciform H-shapes Columns (FCHCs)

**Linfeng Lu** [1,*], **Di Wang** [2], **Zifan Dai** [1], **Tengfei Luo** [1], **Songlin Ding** [1] **and Hanlin Hao** [1]

[1] School of Civil Engineering, Chang'an University, 75 Chang'an Middle Rd, Xi'an 710062, China; 2019228053@chd.edu.cn (Z.D.); 2019228028@chd.edu.cn (T.L.); 2021128017@chd.edu.cn (S.D.); 2021128031@chd.edu.cn (H.H.)

[2] China Construction Third Bureau Group Co., Ltd., Mai Hua Rd, Nanjing 210004, China; wangdi@cscec.com

[*] Correspondence: lulinfeng@chd.edu.cn

**Abstract:** In China, increasing the application ratio of hot-rolled H-shapes has become a severe problem that the government, academia, and engineering circles must vigorously address. Research on reasonable hot-rolled H-shapes built-up columns is one of the primary methods. After reviewing the various combination columns in the existing research, the paper proposes the new flanged cruciform H-shapes columns (FCHCs) made of three hot-rolled H-shapes. Using comprehensive imperfections given by the design standard, GB50017-2017, the paper analyzes the global buckling of FCHCs subjected to the axial compression load. The global buckling factor obtained is compared with the current national design code. Comparative analysis of seventy-two specimens of Q345 and Q460 steel found that the global buckling mode of FCHCs was flexural bending buckling around the axis of symmetry, and global torsional buckling and local buckling did not occur. Furthermore, the corresponding column curves in current design codes overestimate the dimensionless buckling strength of the novel FCHCs. Therefore, designers need to drop a class to select the global buckling factor within a specific range. Finally, new column global buckling curves are proposed based on a non-linear fitting of the numerical results according to the current national design codes.

**Keywords:** hot-rolled shapes; H-shape; flanged cruciform columns; built-up column; global buckling

## 1. Introduction

In many multistory steel moment-resisting frames (SMRFs), the regular column is a universal column (UC) or universal beam (UB), as shown in Figure 1a. In many cases, columns are subjected to both axial and bending effects. Therefore, the designer would control mechanical behavior due to the minor axis problem [1]. According to Chinese GB/T 11263-2017 [2], the ratio of the radius of gyration of UC or UB is from 1.69 to 6.55. The minor axis of a UC or UB usually has a capacity significantly less than its strong axis. Therefore, it leads to an axially loaded column in compression buckles about the minor moment of inertia section.

The method of connecting multiple single components to improve the carrying capacity of the single-member is already a common practice in cold-formed steel structures and aluminum alloy structures [3,4]. A similar approach is also used on hot-rolled steel components. For example, UC or UB are combined with Tee-shapes to form the flanged cruciform columns (FCCs), as shown in Figure 1b, which are suitable for the biaxial bending columns [5]. FCCs have some other names, the cruciform column with universal beam section (CCUB) [6,7], the cruciform column (CC) [8], flanged cruciform sections (FCSs) [9], the stiffened cruciform sections [10], and specially shaped columns with cruciform section [11]. In addition to FCCs, other forms of combinations have also been studied and reported. BWS (Boxed W-shaped section) [12], shown in Figure 1c; SS (side to side) [8], shown in Figure 1d; BBC (built-up battened column) [13], shown in Figure 1e. However, compared to the FCCs, BWS, SS, and BBC often requires internal continuity plates, which is common

to weld at least one side of each plate using electroslag welding (ESW). Still, it is vulnerable to brittle fracture resulting from the notch-like condition created by ESW. The test shows that the bearing capacity of the FCC column with the same cross-sectional area is slightly higher than that of the SS column. [8]. Therefore, FCCs are the better built-up columns.

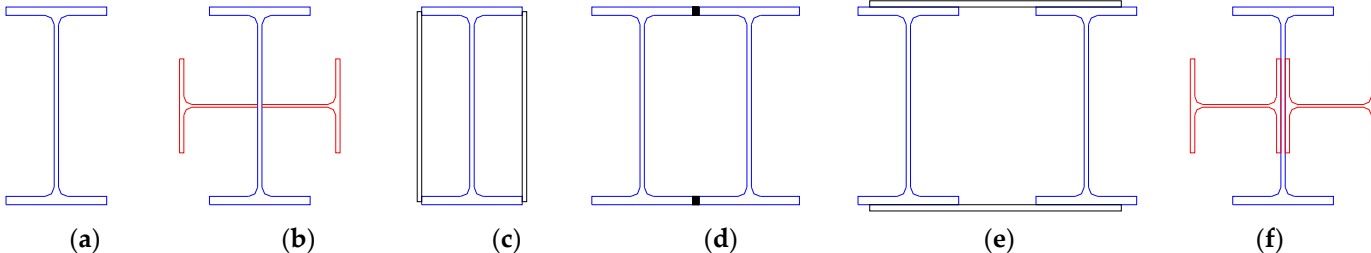

**Figure 1.** The typical of built-up columns with H-shape: (**a**) UC or UB; (**b**) FCC; (**c**) BWS; (**d**) SS; (**e**) BBC; (**f**) FCHC.

The global stability of steel components is an old and young research topic. For instance, some scholars paid attention to the buckling behavior of steel beams and composite beams with special shapes [14,15]. Under normal circumstances, there are generally floor restraints in steel frames to limit the global buckling of the beam. Therefore, the global buckling of the frame column is more important than the beam. Many steel design standards provide comprehensive design aids for analyzing and designing standard steel columns. However, few such design aids are available for built-up sections [16], and many design codes [17,18] do not provide specific design provisions for FCC compression members. Fortunately, some studies focused on or were involved in FCCs [6–11,19]. For example, Tahir. [6] and Hawileh [8] proved that the cruciform column using universal beam sections was an efficient built-up section with a larger ultimate axial load capacity, more significant stiffness, and saved the weight of steel used. Naderian et al. [10] investigated the stability of steel columns with the cruciform section under constant compressive and shear stresses. Zhang et al. [11] conducted the experimental investigation and numerical analysis on the mechanical behavior of FCCs under cyclic loading. Yang et al. [19] studied the instability bearing capacity of the flanged columns with a cruciform section considering residual stress when subjected to axial compression.

At present, China has more than 30 hot-rolled H-shapes production lines, with an annual output of 16 million tons. Still, the application of hot-rolled H-shapes in steel structures is only 15–20%. Therefore, increasing the hot-rolled H-shapes application proportion has been included in the government's work schedule. Based on the combined concept of FCCs, the author directly uses three UC or UB, or three UC and UB to form the flanged cruciform H-shapes columns (FCHCs), as shown in Figure 1f.

Because the global buckling behavior is the base of the axially loaded compression column, this paper conducts numerical investigations on the FCHCs subjected to axial loading. The FE simulation results are used to evaluate its stability behavior and buckling modes and check the applicability of column buckling curves of GB50017-2017 [16], EN 1993-1-1 [17], and ANSI/AISC 360-16 [18].

## 2. Calculation of Geometric Characteristics of the FCHC

Theoretically, the cross-sections of the three hot-rolled H-shapes that make up the FCHC can be completely different. That means designers can choose three different cross-sections or different steel grades of hot-rolled H-beams for combination. However, from the simple design, the frame column section should be biaxially symmetrical or uniaxially symmetrical. Therefore, the other two H-shapes whose flanges are welded to another H-shape are the same section, as shown in Figure 2.

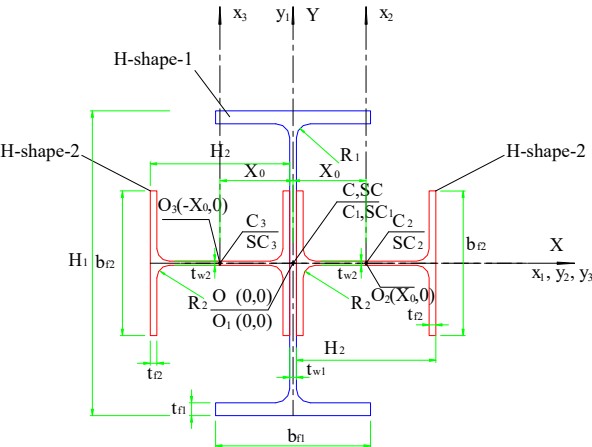

**Figure 2.** Geometric coordinates and dimensions of the FCHC.

The letters C and SC stand for the centroid and shearing center of the H-shape, respectively, as shown in Figure 2, establishing three Cartesian coordinate systems. The whole section coordinate system, XOY, and the local $x_1O_1y_1$ coordinate system are set at H-shape-1 centroid (C1), the local $x_2O_2y_2$ coordinate system is established at the first H-shape-2 centroid (C2), and the local $x_3O_3y_3$ coordinate system is installed at the second H-shape-2 centroid (C3). For the H-shape, the section depth (H), the flange width ($b_f$), the flange thickness ($t_f$), and the web thickness ($t_w$) are also shown in Figure 2.

The subscripts 1 and 2, respectively, represent the H-shape-1 and the H-shape-2. Therefore, under the premise of ignoring fillet welds, in the XOY ($x_1O_1y_1$) coordinate system, the X-axis coordinate value of the origin point $O_2$ ($X_0$,0) and $O_3$ ($-X_0$,0) can be calculated as:

$$X_0 = \frac{H_2}{2} + \frac{t_{w1}}{2}, \tag{1}$$

where $H_2$ is the depth of the H-shape-2; $t_{w1}$ is the web thickness of the H-shape-1.

In the XOY coordinate system, the geometric characteristics of the combined section can be calculated according to the following equations.

$$A = A_1 + 2A_2, \tag{2}$$

$$I_X = I_{x1} + I_{y2} + I_{y3} = I_{x1} + 2I_{y2}, \tag{3}$$

$$I_Y = I_{y1} + 2 \cdot (I_{x2} + A_2 \cdot (X_0)^2), \tag{4}$$

$$i_X = \sqrt{I_X/A}, \tag{5}$$

$$i_Y = \sqrt{I_Y/A}, \tag{6}$$

$$r_0^2 = i_X^2 + i_Y^2 \tag{7}$$

$$J = \sum_{i=1}^{n} \frac{1}{3} b_i t_i^3 \tag{8}$$

$$C_w = 0 \tag{9}$$

$A$ is the cross-sectional area of the combined cross-section, and $A = A_1 + A_2$, $A_1$ and $A_2$ is the cross-sectional area of H-shape-1 and the H-shape-2, respectively.

$I_X$ is the moment of inertia of the combined cross-section about the X-axis under the XOY coordinate system; $I_Y$ is the moment of inertia of the combined cross-section about the Y-axis under the XOY coordinate system.

$i_X$ is the radius of gyration of the combined cross-section about the X-axis; $i_Y$ is the radius of gyration of the combined cross-section about the Y-axis.

$I_{x1}$ and $I_{y1}$ is the moment of inertia of the H-shape-1 about the $x_1$-axis and $y_1$-axis, respectively, under the $x_1O_1y_1$ coordinate system.

$I_{x2}$ and $I_{y2}$ is the moment of inertia of the H-shape-2 about the $x_2$-axis and $y_2$-axis, respectively, under the $x_2O_2y_2$ coordinate system.

$r_o$ is the polar radius of gyration about the shear center.

$J$ is the uniform torsional constant.

$C_w$ is the warping constant.

## 3. Numerical Analysis Method and Verification

### 3.1. Finite Element Model

The author's previous studies [20] successfully used ABAQUS software to perform finite element simulation analysis for steel components. The results show that our finite element modeling method and the selection of analysis units are reasonable and reliable. This research still follows the previous study and uses the same approach to construct the finite element analysis model. In the ABAQUS analysis, eight-node solid non-conforming elements C3D8I were used to reduce grid units and shorten the calculation time. Through trial calculations, it was found that when the mesh size was 20 and 15 mm, the error of the column's stability bearing capacity was less than 0.3%, but the calculation time was shortened by nearly one-third. That shows that when the mesh size is 20 mm, the calculation accuracy can meet the paper's research. Therefore, the maximum mesh size is twenty millimeters in the model (including H-shapes and welds). If the plate thickness of H-shapes and weld leg is less than twenty millimeters, the mesh size takes the actual plate thickness or weld leg size. To build the FE (Finite Element) model, select two kinds of steel recommended in GB 50017-2017 [16], Q345 and Q460. Coupon test results of Q345 steel and Q460 steel are adopted from references [20] and [21], respectively. The stress-strain three-fold line of steels used in the finite element analysis is shown in Figure 3, and the specific data are listed in Table 1. A three-fold line model could consider the strain hardening and descending. The plasticity model is based on the von Mises yielding criteria. Use the "Tie" command to bind the welds and H-shapes. Therefore, only consider the contact between the web of H-shape-1 and the flange of H-shape-2. Set the regular hard contact, and the tangential friction coefficient is set at 0.15. The 3D finite element model of specimens is shown in Figure 4.

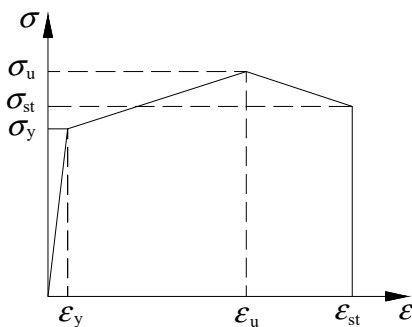

**Figure 3.** Stress-strain relationships: $\sigma_y$ and $\varepsilon_y$ are the yield stress and yield strain, respectively; $\sigma_u$ and $\varepsilon_u$ are the ultimate stress and ultimate strain, respectively; $\sigma_{st}$ and $\varepsilon_{st}$ are the failure stress and failure strain, respectively.

Considering this is the preliminary study, the column's boundary condition is simplified to the pin ended. Thus, the non-loading end of the model is set to pure pin-ended, which can only rotate freely around the *X*-axis and *Y*-axis, and the constraint is $U_X = U_Y = U_Z = 0$. Compared with the non-loading end, the loading end can rotate around the *X*-axis and *Y*-axis and translate along the *X*-axis. That is, the constraint of the *Z*-axis direction needs to be released. The boundary condition is $U_X = U_Y = 0$.

**Table 1.** Material properties of Q345 steel used in ABQUAS.

| | Q345 | | | | | | | Q460 | | | | | | |
|---|---|---|---|---|---|---|---|---|---|---|---|---|---|---|
| Number | $E$/MPa | $\sigma_y$ [1]/MPa | $\varepsilon_y$ | $\sigma_u$/MPa | $\varepsilon_u$ | $\sigma_{st}$/MPa | $\varepsilon_{st}$ | $E$/MPa | $\sigma_y$/MPa | $\varepsilon_y$ | $\sigma_u$/MPa | $\varepsilon_u$ | $\sigma_{st}$/MPa | $\varepsilon_{st}$ |
| 1 | 205,000 | 345 | 0.00184 | 458.55 | 0.11682 | 341.366 | 0.17081 | 210,000 | 531.9 | 0.028 | 657 | 0.140 | 631.3 | 0.2166 |
| 2 | 206,000 | 320 | 0.00164 | 455.83 | 0.13819 | 334.604 | 0.2240 | 212,000 | 492.9 | 0.020 | 643.5 | 0.142 | 598.6 | 0.2382 |
| 3 | 207,000 | 335 | 0.0019 | 461.84 | 0.13457 | 332.958 | 0.20475 | 211,000 | 492.3 | 0.024 | 631.2 | 0.147 | 608.4 | 0.2312 |
| Mean | 206,000 | 333.33 | 0.0018 | 458.33 | 0.1299 | 336.31 | 0.20 | 211,000 | 505.7 | 0.024 | 643.9 | 0.143 | 612.77 | 0.2287 |

[1] Note: $\sigma_y$ is the yield strength and equals $F_y$ as follows.

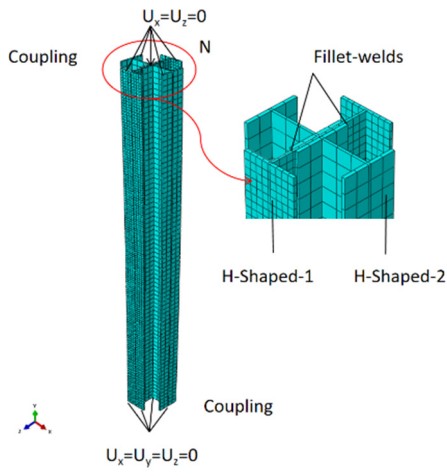

**Figure 4.** The finite element model of FCHCs.

For a two-pin-ended column, the column's effective length is 1.0 $L$, and $L$ is the geometric length of the column. Its calculation diagram is shown in Figure 5. The representative value of the initial imperfections of the column can be calculated and determined according to Table 2; the imperfections value $e_0$ includes the influence of residual stress and other initial geometrical defects. According to GB 50017-2107 [16], the welded FCCs belong to curve b. Therefore, in ABAQUS, the initial imperfections are applied according to curve b in the direction where the column may undergo global flexural buckling (around the minimum radius of gyration).

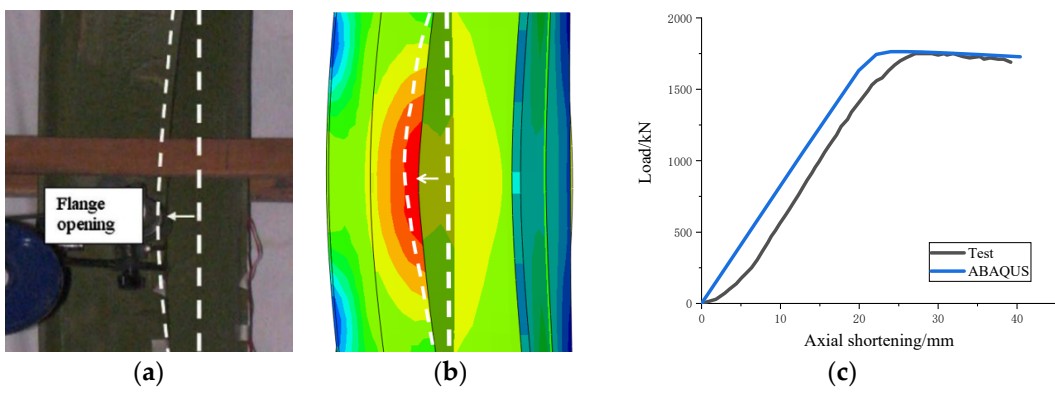

(a)        (b)        (c)

**Figure 5.** Test and ABAQUS analysis of the specimen CCUB1: (**a**) typical failure mode in the test; (**b**) typical failure mode in ABAQUS; (**c**) load-axial shortening curves.

Deniziak and Winkelmann [22] analyzed the efficiency and accuracy of standard FEM (Finite Element Method) calculations performed in ABAQUS software on the example of a critical load assessment of a thin-walled steel column element with selected nonlinearities. According to their studies, we choose CGMNA (Geometrically and Materially Nonlinear Analysis including Contact) in ABAQUS software to check the buckling mode of FCHCs.

In order to realize the CGMNA analysis, non-linear static general analysis is selected in ABAQUS.

**Table 2.** Representative values of comprehensive imperfections of members Table 5.2.2 in GB50017 [16]).

| Column Strength Curves | $e_0/l$ for Second-Order Analysis [2] |
|---|---|
| Curve a | 1/400 |
| Curve b | 1/350 |
| Curve c | 1/300 |
| Curve d | 1/250 |

[2] Note: $e_0$ is the initial deformation value at the midpoint of the component. *l* is the total length of the member.

### 3.2. Verification of ABAQUS Analysis

Literature [6] has reported the axial compression test of FCCs. Therefore, this paper chooses the specimens CCUB1 [6] to verify the ABAQUS numerical analysis.

The global flexural buckling of specimen CCUB1 in the test and ABAQUS are shown in Figure 5. Figure 5 shows that the global flexural buckling mode of ABAQUS is very similar to the experiment. Although the curve indicates that the test and ABAQUS do not overlap, the slopes are the same. The reason is that the systematic displacement (between the test piece and the end sensor and equipment) results in a slip section of the curve, but ABAQUS does not have this slip. Despite this, the test's ultimate loading is 1756 kN, the maximum loading is 1763 kN in ABAQUS, and the error is only 0.39%. That means there is no doubt about the accuracy and reliability of ABAQUS in calculating the ultimate load of the global buckling of the member.

### 4. Design of Analysis Specimens

To check the applicability of compressive strength curves of GB50017-2017 [16], design the columns of each cross-section type according to the pre-controlled slenderness ratio range and divide the columns into short columns ($\lambda \leq 30$), intermediate columns ($30 < \lambda < 100$), and long (slender) columns ($\lambda \geq 100$). The above division is not necessarily very appropriate, but it covers the slenderness ratio range of steel frame columns in engineering design. Select ten H-shapes from GB/T 11263-2017 [2]. Their details are shown in Table 3. Use the ten H-shapes to make FCHCs. The geometric characteristics of all column sections are shown in Table 4. In order to investigate the effect of steel properties, two kinds of structural steel are involved in specimens. Q345 and Q460 are recommended in GB50017-2017 [16]. The sectional geometrical property parameters of FCHCs are calculated by Equations (1)–(6) based on Figure 2. Due to the application of initial comprehensive bending imperfections, the biaxially symmetrical specimen will only undergo global flexural buckling, so the torsional geometric characteristics of the section are not calculated and shown in Table 4.

**Table 3.** Hot-rolled H-shapes for specimens (selected from GB/T 11263-2017 [2]).

| H-Shapes | Section | Geometric Dimensions/mm | | | | | Area/cm$^2$ | Moment of Inertia [3]/cm$^4$ | | Radius of Gyration [3]/cm | |
|---|---|---|---|---|---|---|---|---|---|---|---|
| | | H | B | $T_w$ | $T_f$ | R | | $I_x$ | $I_y$ | $I_x$ | $I_y$ |
| HW | HW 150 * 150 | 150 | 150 | 7 | 10 | 8 | 30 | 1620 | 563 | 6.39 | 3.76 |
| | HW 200 * 200 | 200 | 200 | 8 | 12 | 13 | 63.53 | 4720 | 1600 | 8.61 | 5.02 |
| | HW 250 * 250 | 250 | 250 | 9 | 14 | 13 | 91.43 | 10700 | 3650 | 10.8 | 6.31 |
| HM | HM 300 * 200 | 294 | 200 | 8 | 12 | 13 | 71.05 | 11100 | 1600 | 12.5 | 4.74 |
| | HM 350 * 250 | 340 | 250 | 9 | 14 | 13 | 99.53 | 21200 | 3650 | 14.6 | 6.05 |
| | HM 400 * 300 | 390 | 300 | 10 | 16 | 13 | 133.3 | 37900 | 7200 | 16.9 | 7.35 |
| HN | HN 300 * 150 | 300 | 150 | 6.5 | 9 | 13 | 46.78 | 7210 | 508 | 12.4 | 3.29 |
| | HN 400 * 200 | 400 | 200 | 8 | 13 | 13 | 83.37 | 23500 | 1740 | 16.8 | 4.56 |
| | HN 450 * 200 | 450 | 200 | 9 | 14 | 13 | 95.43 | 32900 | 1870 | 18.6 | 4.42 |
| | HN 500 * 200 | 500 | 200 | 10 | 16 | 13 | 112.3 | 46800 | 2140 | 20.4 | 4.36 |

[3] Note: The subscripts x and y indicate the major axis and minor axis of a single H-shape.

**Table 4.** Details of FCHCs.

| Specimen | Group [4] | Section | Area/cm$^2$ | Moment of Inertia/cm$^4$ | | Radius of Gyration/cm | | $\lambda = L/\min(i_x, i_y)$ | | | | $\varepsilon_k = \sqrt{235/f_y}$ | |
|---|---|---|---|---|---|---|---|---|---|---|---|---|---|
| | | | | $I_X$ | $I_Y$ | $i_X$ | $i_Y$ | L = 12m | L = 9m | L = 6m | L = 3m | Q345 | Q460 |
| FCHC-X-1 | H-shape-1<br>H-shape-2 | HN 300 * 150<br>HN 150 * 150 | 106.78 | 8336 | 7422 | 8.84 | 8.34 | 143.88 | 107.91 | 71.94 | 35.97 | 0.825 | 0.715 |
| FCHC-X-2 | H-shape-1<br>H-shape-2 | HN 400 * 200<br>HN 200 * 200 | 210.43 | 26,700 | 24,923 | 11.26 | 10.88 | 110.29 | 82.72 | 55.15 | 27.57 | 0.825 | 0.715 |
| FCHC-X-3 | H-shape-1<br>H-shape-2 | HN 450 * 200<br>HN 250 * 250 | 278.29 | 40,200 | 53,936 | 12.02 | 13.92 | 99.83 | 74.88 | 49.92 | 24.96 | 0.825 | 0.715 |
| FCHC-X-4 | H-shape-1<br>H-shape-2 | HN 500 * 200<br>HN 250 * 250 | 295.16 | 54,100 | 54,443 | 13.54 | 13.58 | 88.63 | 66.47 | 44.31 | 22.16 | 0.825 | 0.715 |
| FCHC-X-5 | H-shape-1<br>H-shape-2 | HM 300 * 200<br>HW150 * 150 | 131.05 | 12,226 | 8585 | 9.66 | 8.09 | 148.33 | 111.25 | 74.17 | 37.08 | 0.825 | 0.715 |
| FCHC-X-6 | H-shape-1<br>H-shape-2 | HM 350 * 250<br>HW 200 * 200 | 226.59 | 24,400 | 26,965 | 10.38 | 10.91 | 115.61 | 86.71 | 57.80 | 28.90 | 0.825 | 0.715 |
| FCHC-X-7 | H-shape-1<br>H-shape-2 | HM 400 * 300<br>HM 200 * 200 | 260.36 | 41,100 | 30,648 | 12.56 | 10.85 | 110.60 | 82.95 | 55.30 | 27.65 | 0.825 | 0.715 |
| FCHC-X-8 | H-shape-1<br>H-shape-2 | HM 500 * 200<br>HW 300 * 200 | 254.4 | 50,000 | 57,171 | 14.02 | 14.99 | 85.59 | 64.19 | 42.80 | 21.40 | 0.825 | 0.715 |
| FCHC-X-9 | H-shape-1<br>H-shape-2 | HM 500 * 200<br>HM 300 * 150 | 205.86 | 47,816 | 39,038 | 15.24 | 13.77 | 87.15 | 65.36 | 43.57 | 21.79 | 0.825 | 0.715 |

[4] Note: X is 1 and 2, according to Q345 and Q460. The $\lambda$ is calculated according to Q235. The subscripts X and Y indicate the principal axes (*X*-axis and *Y*-axis) of the FCHCs shown in Figure 2.

## 5. Results, Observations, and Discussion

### 5.1. Load-Deformation Curves

As shown from Table 4, FCHC-X-1, 2, 5, 7, and 9 group specimens will occur global flexural buckling about the *Y*-axis ($i_Y < i_X$); the others will buckle about the *X*-axis ($i_X < i_Y$). Therefore, two series of models, FCHC-X-3 and FCHC-X-7, are selected as representatives from the two types of global buckling mode specimen groups to illustrate the typical buckling modes. Figure 6 demonstrates that the load-deformation curves of the FCHC-X-3 series specimens and the FCHC-X-7 series specimens. The deformation is the horizontal displacement of the column mid-length point along with the flexural buckling direction.

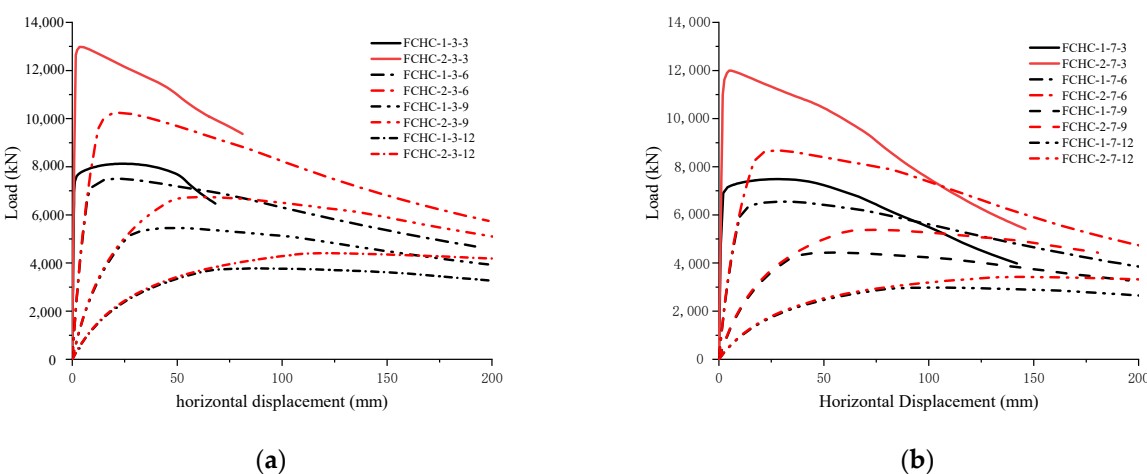

**Figure 6.** Loading-deformation curves: (**a**) FCHC-X-1 series; (**b**) FCHC-X-7 series.

It can be seen from Figure 6 that the Q345 specimen has the same global buckling mode as the Q460 specimen with the same length except for a higher buckling capacity. The lateral deformation of the sample with a height of three meters is minimal, which somewhat shows the failure characteristics of a short column under axial compression, while the specimens of other lengths are more in line with the features of the globe buckling of intermediate columns.

### 5.2. Buckling Mode and Buckling Deformation

FCHC-X-3 and FCHC-X-7 series specimens are randomly selected as representative specimens of different buckling directions (*X*-axis or *Y*-axis). The typical buckling modes of two series specimens are shown in Figures 7–10. Their global buckling mode diagram and the deformation diagram at the mid-length section during buckling (critical state) observe whether the global buckling is related to the local buckling.

Figures 7–10 illustrate that the global buckling mode of all specimens is not related to local buckling, and no local buckling of the plate occurs in all models. At the same time, it also shows that the local stability of the hot-rolled H-shapes in FCHCs meets the limit of the width-to-thickness ratio or the depth-to-thickness ratio in the design standard GB50017 as always.

From Equation (9), the warping constant $C_w = 0$. Therefore, it is easy for the global torsional buckling for an ideal Euler FCHC to occur, especially for a short column or an intermediate column. However, under an inevitable initial imperfection, the global buckling mode of FCHCs is flexural buckling and not torsional buckling. As the columns of the steel moment-resisting frames are always the compression-bending members (beam-columns), global torsional buckling never happens.

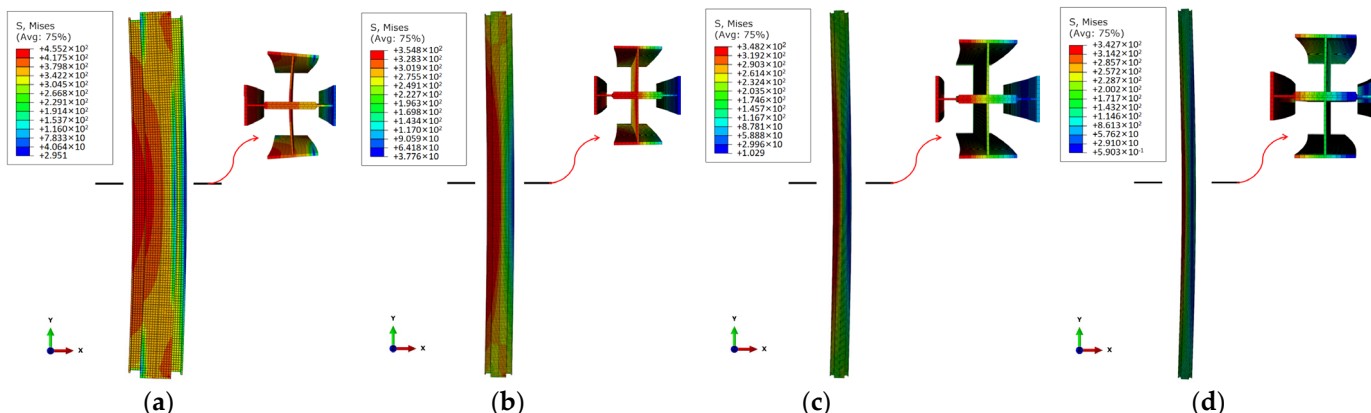

**Figure 7.** The FCHC-1-3 series illustrates the typical failure mode of the flexural-torsional buckling: (**a**) FCHC-1-3-3; (**b**) FCHC-1-3-6; (**c**) FCHC-1-3-9; (**d**) FCHC-1-3-12.

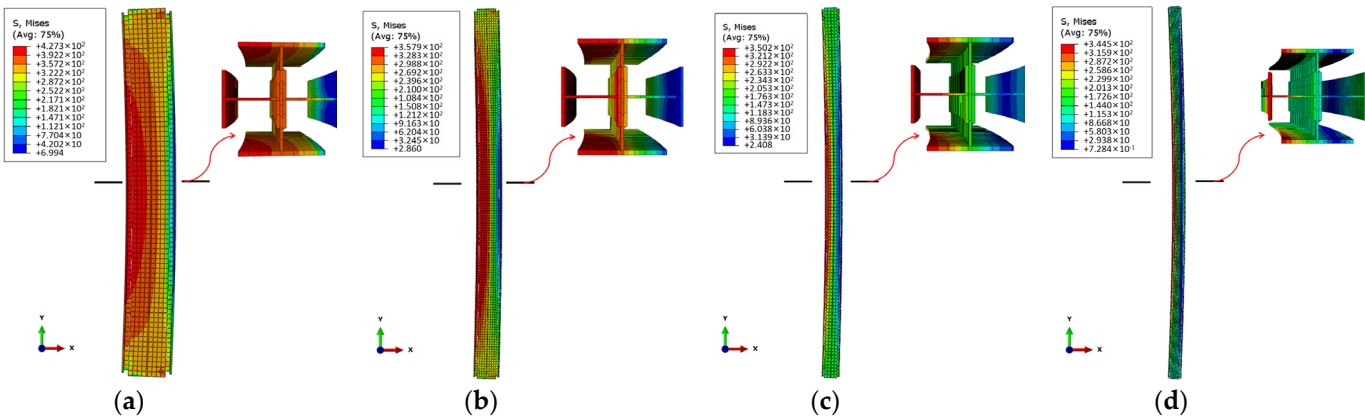

**Figure 8.** The FCHC-1-7 series illustrates the typical failure mode of the flexural-torsional buckling: (**a**) FCHC-1-7-3; (**b**) FCHC-1-7-6; (**c**) FCHC-1-7-9; (**d**) FCHC-1-7-12.

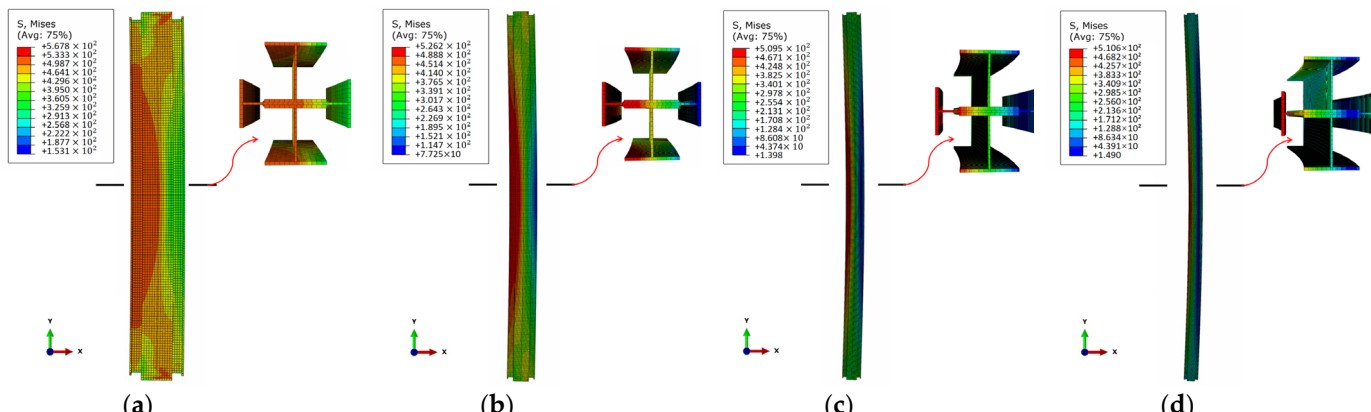

**Figure 9.** The FCHC-2-3 series illustrates the typical failure mode of the flexural–torsional buckling: (**a**) FCHC-2-3-3; (**b**) FCHC-2-3-6; (**c**) FCHC-2-3-9; (**d**) FCHC-2-3-12.

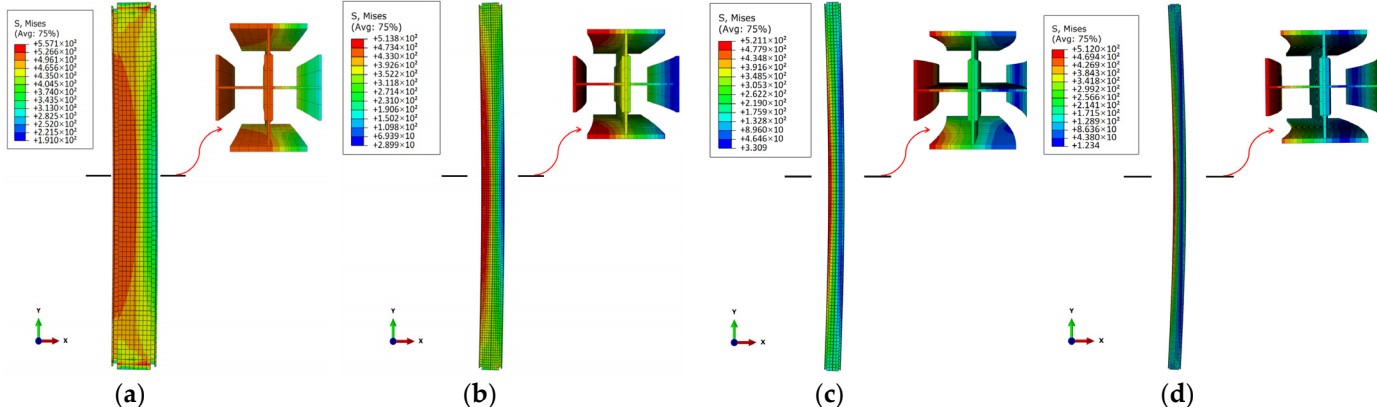

**Figure 10.** The FCHC-2-7 series illustrates the typical failure mode of the flexural buckling: (**a**) FCHC-2-7-3; (**b**) FCHC-2-7-6; (**c**) FCHC-2-7-9; (**d**) FCHC-2-7-12.

The stability analysis of the axial compression column is the basis of the overall stability design of the compression-bending column, especially the global flexural stability factor of the axial compression column is the critical parameter in the design relevant equation.

### 5.3. Global Flexural Buckling Factors in Current Design Codes

In GB50017-2017 [16], the global flexural buckling factor $\varphi$ of the axially loaded compression member should be calculated according to the following formula.

When $\lambda_n \leq 0.215$,

$$\varphi = 1 - \alpha_1 \lambda_n^2, \tag{10}$$

When $\lambda_n > 0.215$,

$$\varphi = \frac{1}{2\lambda_n^2}\left[\left(\alpha_2 + \alpha_3\lambda_n + \lambda_n^2\right) - \sqrt{\left(\alpha_2 + \alpha_3\lambda_n + \lambda_n^2\right)^2 - 4\lambda_n^2}\right], \tag{11}$$

where $\lambda_n$ is dimensionless slenderness ratio, $\lambda_n = \frac{\lambda}{\pi}\sqrt{f_y/E}$. $\alpha_1$, $\alpha2$, and $\alpha_3$ is the coefficient, classified according to the section classification and adapted according to Table 5.

**Table 5.** Coefficient $\alpha1$, $\alpha2$, and $\alpha3$ according to GB 50017-2017 [16].

| Column Buckling Curve | | $\alpha_1$ | $\alpha_2$ | $\alpha_3$ |
|---|---|---|---|---|
| a | | 0.41 | 0.986 | 0.152 |
| b | | 0.65 | 0.965 | 0.300 |
| c | $\lambda_n \leq 1.05$ | 0.73 | 0.906 | 0.595 |
| | $\lambda_n > 1.05$ | | 1.216 | 0.302 |
| d | $\lambda_n \leq 1.05$ | 1.35 | 0.868 | 0.915 |
| | $\lambda_n > 1.05$ | | 1.375 | 0.432 |

In EN 1993-1-1 [17], the global flexural buckling curve of the axially loaded compression member is expressed according to the following.

$$\varphi = \frac{1}{\Phi + \sqrt{\Phi^2 - \lambda_n^2}} \text{ but } \varphi \leq 1.0, \tag{12}$$

where, $\Phi = 0.5[1 + \alpha(\lambda_n - 0.2) + \lambda_n^2]$, $\alpha$ is an imperfection factor corresponding to the appropriate buckling curve should be obtained from Table 6.

**Table 6.** Imperfection factors for buckling curves in EN 1993-1-1 [17].

| Column Buckling Curve | $a_0$ | a | b | c | d |
|---|---|---|---|---|---|
| Imperfection factor $\alpha$ | 0.13 | 0.21 | 0.34 | 0.49 | 0.76 |

ANSI/AISC 360-16 [18] only has one column buckling strength curve, expressed according to the following.

When $\lambda_n \leq 1.5$,

$$\varphi = 0.658^{\frac{F_y}{F_e}}, \tag{13}$$

When $\lambda_n > 1.5$,

$$\varphi = \frac{0.877 F_e}{F_y}, \tag{14}$$

$F_e = \frac{\pi^2 E}{\lambda^2}$, $E$ is the modulus of elasticity of steel; $F_y$ is the specified minimum yield stress of the type of steel being used (MPa).

The buckling capacity (the peak load $N_u$), global flexural buckling factor $\varphi$ ($\varphi = N_u/(f_y \cdot A)$), slenderness ratio $\lambda$ ($\lambda = L_c/i$), and dimensionless slenderness ratio $\lambda_n$ ($\lambda_n = \frac{\lambda}{\pi}\sqrt{f_y/E}$) were obtained and summarized in Tables 7 and 8.

It should be noted that, in Tables 7 and 8, the letters b and c of Curve b and Curve c represent the column buckling curves of type of b and c according to cross-section classifications taken from GB50017 and EN1993-1-1, respectively.

For all Q345 specimens, the ABAQUS results ($\varphi^A$) are lower than the corresponding design curve (i.e., curve b) in both GB 50,017 (Figure 11a) and EN 1993-1-1 (Figure 11b) by an average of −1.2 and −1.4%, respectively, and higher than the curve c in GB 50,017 by 11.5% and EN 1993-1-1 by 5.9%. The FE results ($\varphi^A$) are also lower than the design curve in ANSI/AISC 360 (Figure 11c) by −7.5% on average. However, for the Q460 specimen, the $\varphi^A$ is lower than the corresponding design curve b in both GB 50,017 (Figure 12a) and EN 1993-1-1 (Figure 12b) by an average of −0.9 and −0.8%, respectively, and higher than the curve c in GB 50,017 by 11.8% and EN 1993-1-1 by 6.8%. In addition, the $\varphi^A$ is lower than the design curve in ANSI/AISC 360 (Figure 12c) by −7.5% on average. Thus, the design provision in ANSI/AISC 360-16 [18] is unsuitable for the global buckling design of FCHCs.

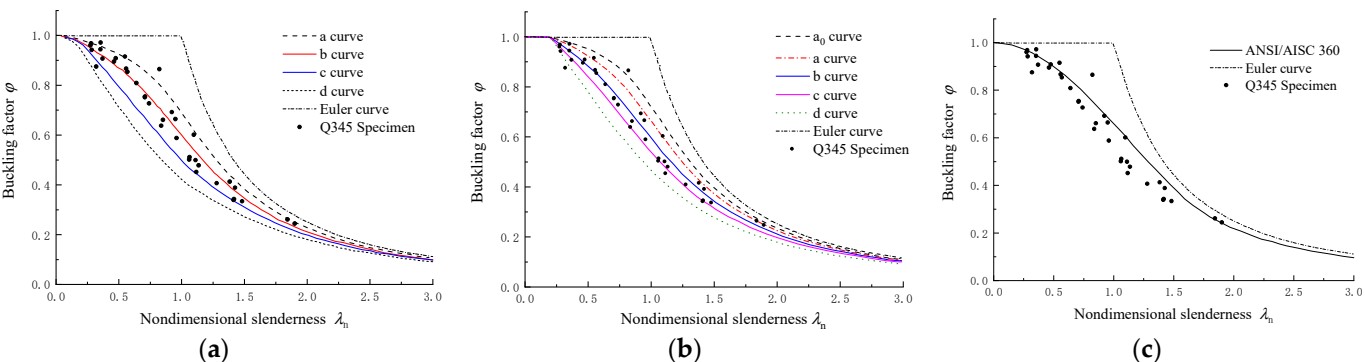

**Figure 11.** Comparison of FE results with design curves for Q345 specimens: (**a**) GB50017; (**b**) EN1993-1-1; (**c**) ANSI/AISC360.

**Table 7.** Buckling resistance for Q345 specimens with comprehensive imperfections of Curve b in Table 2.

| Specimen [5] | Buckling Load $N_u$/kN | $\varphi^A = \frac{N_u}{A \cdot f_y}$ | $\lambda$ | $\lambda_n$ | GB 50017 | | EN 1993-1-1 | | ANSI/AISC 360 | Errors of GB 50017/% | | Errors of EN 1993-1-1/% | | Errors of ANSI/AISC 360/% |
|---|---|---|---|---|---|---|---|---|---|---|---|---|---|---|
| | | | | | $\varphi^b$ | $\varphi^c$ | $\varphi^b$ | $\varphi^c$ | $\varphi_A$ | Curve b | Curve c | Curve b | Curve c | |
| FCHC-1-1-3 | 3185.85 | 0.895 | 35.97 | 0.46 | 0.887 | 0.821 | 0.901 | 0.865 | 0.915 | 0.9 | 9.0 | −0.7 | 3.5 | −2.2 |
| FCHC-1-1-6 | 2464.14 | 0.692 | 71.94 | 0.92 | 0.650 | 0.543 | 0.647 | 0.587 | 0.701 | 6.5 | 27.6 | 7.0 | 18.0 | −1.2 |
| FCHC-1-1-9 | 1470.55 | 0.413 | 107.91 | 1.38 | 0.394 | 0.347 | 0.389 | 0.356 | 0.449 | 4.8 | 19.0 | 6.2 | 16.1 | −8.1 |
| FCHC-1-1-12 | 930.98 | 0.262 | 143.88 | 1.84 | 0.245 | 0.227 | 0.242 | 0.225 | 0.258 | 6.8 | 15.0 | 8.1 | 16.1 | 1.3 |
| FCHC-1-2-3 | 6624.04 | 0.944 | 27.57 | 0.35 | 0.926 | 0.885 | 0.944 | 0.922 | 0.949 | 2.0 | 6.8 | 0.0 | 2.4 | −0.5 |
| FCHC-1-2-6 | 5274.34 | 0.752 | 55.15 | 0.71 | 0.776 | 0.671 | 0.780 | 0.721 | 0.811 | −3.1 | 12.1 | −3.6 | 4.3 | −7.3 |
| FCHC-1-2-9 | 3518.71 | 0.502 | 82.72 | 1.06 | 0.564 | 0.469 | 0.560 | 0.506 | 0.625 | −11.1 | 7.0 | −10.4 | −0.9 | −19.7 |
| FCHC-1-2-12 | 2383.75 | 0.340 | 110.29 | 1.41 | 0.381 | 0.337 | 0.376 | 0.344 | 0.434 | −10.8 | 0.8 | −9.7 | −1.3 | −21.6 |
| FCHC-1-3-3 | 8118.69 | 0.875 | 24.96 | 0.32 | 0.937 | 0.904 | 0.957 | 0.939 | 0.958 | −6.6 | −3.2 | −8.5 | −6.8 | −8.7 |
| FCHC-1-3-6 | 7504.94 | 0.809 | 49.92 | 0.64 | 0.810 | 0.712 | 0.817 | 0.762 | 0.843 | −0.1 | 13.6 | −0.9 | 6.2 | −4.0 |
| FCHC-1-3-9 | 5455.71 | 0.588 | 74.88 | 0.96 | 0.626 | 0.522 | 0.623 | 0.564 | 0.680 | −6.1 | 12.7 | −5.6 | 4.3 | −13.6 |
| FCHC-1-3-12 | 3775.76 | 0.407 | 99.83 | 1.28 | 0.442 | 0.383 | 0.437 | 0.398 | 0.504 | −8.0 | 6.3 | −6.9 | 2.3 | −19.3 |
| FCHC-1-4-3 | 9268.63 | 0.942 | 22.16 | 0.28 | 0.948 | 0.925 | 0.970 | 0.957 | 0.967 | −0.7 | 1.8 | −2.9 | −1.6 | −2.6 |
| FCHC-1-4-6 | 8402.15 | 0.854 | 44.31 | 0.57 | 0.843 | 0.756 | 0.853 | 0.804 | 0.874 | 1.3 | 12.9 | 0.1 | 6.2 | −2.3 |
| FCHC-1-4-9 | 6506.96 | 0.661 | 66.47 | 0.85 | 0.693 | 0.583 | 0.692 | 0.630 | 0.738 | −4.6 | 13.4 | −4.4 | 5.0 | −10.4 |
| FCHC-1-4-12 | 4708.24 | 0.479 | 88.63 | 1.14 | 0.519 | 0.432 | 0.514 | 0.466 | 0.583 | −7.9 | 10.7 | −7.0 | 2.7 | −17.9 |
| FCHC-1-5-3 | 3967.94 | 0.908 | 37.08 | 0.48 | 0.882 | 0.812 | 0.895 | 0.857 | 0.910 | 3.0 | 11.8 | 1.5 | 6.0 | −0.2 |
| FCHC-1-5-6 | 2902.73 | 0.665 | 74.17 | 0.95 | 0.632 | 0.527 | 0.629 | 0.569 | 0.685 | 5.1 | 26.2 | 5.7 | 16. | −3.0 |
| FCHC-1-5-9 | 1698.62 | 0.389 | 111.25 | 1.43 | 0.376 | 0.333 | 0.371 | 0.340 | 0.427 | 3.4 | 16.7 | 4.7 | 14.4 | −9.0 |
| FCHC-1-5-12 | 1068.43 | 0.245 | 148.33 | 1.90 | 0.232 | 0.216 | 0.229 | 0.214 | 0.243 | 5.4 | 13.0 | 6.7 | 14.2 | 0.7 |
| FCHC-1-6-3 | 6852.63 | 0.907 | 28.9 | 0.37 | 0.920 | 0.875 | 0.938 | 0.913 | 0.944 | −1.4 | 3.7 | −3.2 | −0.6 | −3.9 |
| FCHC-1-6-6 | 5494.11 | 0.727 | 57.8 | 0.74 | 0.758 | 0.650 | 0.760 | 0.700 | 0.795 | −4.0 | 11.9 | −4.3 | 4.0 | −8.5 |
| FCHC-1-6-9 | 3773.66 | 0.500 | 86.71 | 1.11 | 0.534 | 0.448 | 0.529 | 0.479 | 0.597 | −6.4 | 11.6 | −5.5 | 4.4 | −16.3 |
| FCHC-1-6-12 | 2524.81 | 0.334 | 115.61 | 1.48 | 0.354 | 0.316 | 0.349 | 0.321 | 0.399 | −5.5 | 5.7 | −4.3 | 4.2 | −16.3 |
| FCHC-1-7-3 | 7486.38 | 0.863 | 27.65 | 0.35 | 0.925 | 0.884 | 0.944 | 0.921 | 0.949 | −6.8 | −2.4 | −8.6 | −6.4 | −9.1 |
| FCHC-1-7-6 | 6551.61 | 0.755 | 55.3 | 0.71 | 0.775 | 0.670 | 0.779 | 0.719 | 0.811 | −2.6 | 12.7 | −3.1 | 4.9 | −6.9 |
| FCHC-1-7-9 | 4437.98 | 0.511 | 82.95 | 1.06 | 0.562 | 0.467 | 0.558 | 0.504 | 0.623 | −9.1 | 9.4 | −8.3 | 1.4 | −18.0 |
| FCHC-1-7-12 | 2980.45 | 0.343 | 110.6 | 1.42 | 0.379 | 0.336 | 0.375 | 0.343 | 0.432 | −9.5 | 2.2 | −8.3 | 0.1 | −20.4 |

**Table 7.** *Cont.*

| Specimen [5] | Buckling Load $N_u$/kN | $\varphi^A = \frac{N_u}{A \cdot f_y}$ | $\lambda$ | $\lambda_n$ | GB 50017 | | EN 1993-1-1 | | ANSI/AISC 360 | Errors of GB 50017/% | | Errors of EN 1993-1-1/% | | Errors of ANSI/AISC 360/% |
|---|---|---|---|---|---|---|---|---|---|---|---|---|---|---|
| | | | | | $\varphi^b$ | $\varphi^c$ | $\varphi^b$ | $\varphi^c$ | $\varphi_A$ | Curve b | Curve c | Curve b | Curve c | |
| FCHC-1-8-3 | 8141.4 | 0.960 | 21.4 | 0.27 | 0.952 | 0.931 | 0.974 | 0.962 | 0.969 | 0.9 | 3.1 | −1.4 | −0.2 | −0.9 |
| FCHC-1-8-6 | 7761.5 | 0.915 | 42.8 | 0.55 | 0.852 | 0.768 | 0.862 | 0.816 | 0.882 | 7.5 | 19.2 | 6.2 | 12.2 | 3.8 |
| FCHC-1-8-9 | 7332.14 | 0.865 | 64.19 | 0.82 | 0.711 | 0.600 | 0.711 | 0.648 | 0.753 | 21.7 | 44.0 | 21.7 | 33.4 | 14.8 |
| FCHC-1-8-12 | 5100.31 | 0.601 | 85.59 | 1.10 | 0.542 | 0.454 | 0.537 | 0.486 | 0.605 | 10.9 | 32.6 | 11.9 | 23.7 | −0.5 |
| FCHC-1-9-3 | 6646.77 | 0.969 | 21.79 | 0.28 | 0.950 | 0.928 | 0.972 | 0.960 | 0.968 | 2.0 | 4.4 | −0.3 | 0.9 | 0.1 |
| FCHC-1-9-6 | 5947.22 | 0.867 | 43.57 | 0.56 | 0.847 | 0.762 | 0.858 | 0.810 | 0.878 | 2.3 | 13.7 | 1.1 | 7.0 | −1.3 |
| FCHC-1-9-9 | 4375.57 | 0.638 | 65.36 | 0.84 | 0.702 | 0.592 | 0.701 | 0.639 | 0.746 | −9.1 | 7.8 | −9.1 | −0.2 | −14.5 |
| FCHC-1-9-12 | 3100.73 | 0.452 | 87.15 | 1.12 | 0.530 | 0.446 | 0.525 | 0.476 | 0.594 | −14.8 | 1.4 | −14.0 | −5.0 | −23.9 |
| Average | – | – | – | – | – | – | – | – | – | −1.2 | 11.5 | −1.4 | 5.9 | −7.5 |

[5] Note: For example, FCHC-1-1-3 is represented as the first group of specimens in Table 4. The material of the specimen is Q345, and the effective length is 3 m.

**Table 8.** Buckling resistance for Q460 specimens with comprehensive imperfections of Curve b in Table 2.

| Specimen [6] | Buckling Load $N_u$/kN | $\varphi^A = \frac{N_u}{A \cdot f_y}$ | $\lambda$ | $\lambda_n$ | GB 50017 | | EN 1993-1-1 | | ANSI/AISC 360 | Errors of GB 50017% | | Errors of EN 1993-1-1% | | Errors of ANSI/AISC 360% |
|---|---|---|---|---|---|---|---|---|---|---|---|---|---|---|
| | | | | | $\varphi^b$ | $\varphi^c$ | $\varphi^b$ | $\varphi^c$ | $\varphi_A$ | Curve b | Curve c | Curve b | Curve c | |
| FCHC-2-1-3 | 4823.37 | 0.893 | 35.97 | 0.56 | 0.846 | 0.760 | 0.856 | 0.808 | 0.877 | 5.5 | 17.5 | 4.3 | 10.5 | 1.9 |
| FCHC-2-1-6 | 3096.42 | 0.573 | 71.94 | 1.12 | 0.527 | 0.439 | 0.522 | 0.473 | 0.591 | 8.7 | 30.7 | 9.8 | 21.3 | −2.9 |
| FCHC-2-1-9 | 1702.21 | 0.315 | 107.91 | 1.68 | 0.287 | 0.262 | 0.283 | 0.262 | 0.310 | 10.0 | 20.2 | 11.4 | 20.3 | 1.7 |
| FCHC-2-1-12 | 1036.94 | 0.192 | 143.88 | 2.24 | 0.172 | 0.164 | 0.170 | 0.161 | 0.174 | 11.4 | 17.2 | 12.7 | 19.5 | 10.2 |
| FCHC-2-2-3 | 9668.4 | 0.909 | 27.57 | 0.43 | 0.899 | 0.839 | 0.914 | 0.881 | 0.926 | 1.1 | 8.3 | −0.6 | 3.1 | −1.8 |
| FCHC-2-2-6 | 6877.06 | 0.646 | 55.15 | 0.86 | 0.688 | 0.578 | 0.687 | 0.625 | 0.734 | −6.1 | 11.8 | −5.9 | 3.5 | −11.9 |
| FCHC-2-2-9 | 4299.73 | 0.404 | 82.72 | 1.29 | 0.437 | 0.367 | 0.432 | 0.393 | 0.498 | −7.5 | 10.2 | −6.4 | 2.8 | −18.9 |
| FCHC-2-2-12 | 2755.97 | 0.259 | 110.29 | 1.72 | 0.276 | 0.254 | 0.273 | 0.253 | 0.297 | −6.2 | 2.1 | −5.0 | 2.4 | −12.7 |
| FCHC-2-3-3 | 13,277.21 | 0.922 | 24.96 | 0.39 | 0.913 | 0.863 | 0.930 | 0.903 | 0.939 | 1.0 | 6.8 | −0.9 | 2.1 | −1.7 |
| FCHC-2-3-6 | 11,027.1 | 0.728 | 49.92 | 0.78 | 0.736 | 0.627 | 0.738 | 0.676 | 0.776 | −1.1 | 16.1 | −1.3 | 7.7 | −6.2 |
| FCHC-2-3-9 | 7836.95 | 0.479 | 74.88 | 1.17 | 0.501 | 0.418 | 0.496 | 0.450 | 0.565 | −4.5 | 14.7 | −3.5 | 6.5 | −15.3 |
| FCHC-2-3-12 | 5427.25 | 0.313 | 99.83 | 1.56 | 0.326 | 0.295 | 0.322 | 0.297 | 0.362 | −4.0 | 6.4 | −2.8 | 5.5 | −13.4 |
| FCHC-2-4-3 | 14,044.01 | 0.911 | 22.16 | 0.35 | 0.928 | 0.889 | 0.947 | 0.926 | 0.951 | −1.9 | 2.4 | −3.9 | −1.6 | −4.3 |
| FCHC-2-4-6 | 11,477.5 | 0.782 | 44.31 | 0.69 | 0.784 | 0.680 | 0.789 | 0.730 | 0.819 | −0.3 | 14.9 | −0.8 | 7.1 | −4.5 |
| FCHC-2-4-9 | 8021.87 | 0.555 | 66.47 | 1.04 | 0.579 | 0.481 | 0.574 | 0.519 | 0.638 | −4.1 | 15.4 | −3.4 | 6.9 | −13.0 |

Table 8. *Cont.*

| Specimen [6] | Buckling Load $N_u$/kN | $\varphi^A=\frac{N_u}{A\cdot f_y}$ | $\lambda$ | $\lambda_n$ | GB 50017 | | EN 1993-1-1 | | ANSI/AISC 360 | Errors of GB 50017% | | Errors of EN 1993-1-1% | | Errors of ANSI/AISC 360% |
|---|---|---|---|---|---|---|---|---|---|---|---|---|---|---|
| | | | | | $\varphi^b$ | $\varphi^c$ | $\varphi^b$ | $\varphi^c$ | $\varphi_A$ | Curve b | Curve c | Curve b | Curve c | |
| FCHC-2-4-12 | 5513.34 | 0.377 | 88.63 | 1.38 | 0.394 | 0.333 | 0.389 | 0.356 | 0.450 | −4.3 | 13.3 | −3.1 | 6.0 | −16.1 |
| FCHC-2-5-3 | 5455.35 | 0.823 | 37.08 | 0.58 | 0.839 | 0.750 | 0.848 | 0.798 | 0.869 | −1.8 | 9.8 | −2.9 | 3.1 | −5.3 |
| FCHC-2-5-6 | 3605.41 | 0.544 | 74.17 | 1.16 | 0.507 | 0.423 | 0.502 | 0.455 | 0.571 | 7.2 | 28.7 | 8.3 | 19.5 | −4.8 |
| FCHC-2-5-9 | 1954.77 | 0.295 | 111.25 | 1.73 | 0.272 | 0.250 | 0.269 | 0.249 | 0.292 | 8.4 | 17.8 | 9.8 | 18.3 | 1.2 |
| FCHC-2-5-12 | 1185.65 | 0.179 | 148.33 | 2.31 | 0.163 | 0.155 | 0.161 | 0.152 | 0.164 | 9.8 | 15.1 | 11.0 | 17.5 | 9.1 |
| FCHC-2-6-3 | 10,438.2 | 0.894 | 28.9 | 0.45 | 0.891 | 0.827 | 0.905 | 0.870 | 0.919 | 0.3 | 8.1 | −1.3 | 2.7 | −2.7 |
| FCHC-2-6-6 | 7526.8 | 0.653 | 57.8 | 0.90 | 0.663 | 0.554 | 0.660 | 0.599 | 0.712 | −1.5 | 17.8 | −1.1 | 9.0 | −8.3 |
| FCHC-2-6-9 | 4680.68 | 0.400 | 86.71 | 1.35 | 0.408 | 0.357 | 0.403 | 0.368 | 0.465 | −1.9 | 11.9 | −0.7 | 8.7 | −14.1 |
| FCHC-2-6-12 | 2993.16 | 0.251 | 115.61 | 1.80 | 0.255 | 0.236 | 0.251 | 0.234 | 0.270 | −1.3 | 6.6 | −0.1 | 7.4 | −6.9 |
| FCHC-2-7-3 | 11,999.7 | 0.911 | 27.65 | 0.43 | 0.898 | 0.839 | 0.914 | 0.881 | 0.925 | 1.5 | 8.7 | −0.2 | 3.5 | −1.5 |
| FCHC-2-7-6 | 8685.63 | 0.660 | 55.3 | 0.86 | 0.686 | 0.577 | 0.685 | 0.623 | 0.733 | −3.9 | 14.4 | −3.8 | 5.8 | −10.0 |
| FCHC-2-7-9 | 5378.89 | 0.409 | 82.95 | 1.29 | 0.435 | 0.365 | 0.430 | 0.392 | 0.497 | −6.1 | 11.8 | −5.0 | 4.3 | −17.7 |
| FCHC-2-7-12 | 3421.55 | 0.260 | 110.6 | 1.72 | 0.275 | 0.253 | 0.271 | 0.252 | 0.295 | −5.4 | 2.9 | −4.2 | 3.2 | −11.9 |
| FCHC-2-8-3 | 12,190.7 | 0.928 | 21.4 | 0.33 | 0.932 | 0.896 | 0.952 | 0.932 | 0.954 | −0.4 | 3.6 | −2.5 | −0.4 | −2.8 |
| FCHC-2-8-6 | 10,538.5 | 0.782 | 42.8 | 0.67 | 0.796 | 0.695 | 0.802 | 0.745 | 0.830 | −1.8 | 12.5 | −2.5 | 5.0 | −5.8 |
| FCHC-2-8-9 | 7597.49 | 0.570 | 64.19 | 1.00 | 0.600 | 0.499 | 0.597 | 0.539 | 0.658 | −5.1 | 14.2 | −4.5 | 5.6 | −13.3 |
| FCHC-2-8-12 | 5380.87 | 0.396 | 85.59 | 1.33 | 0.416 | 0.363 | 0.411 | 0.375 | 0.475 | −4.6 | 9.1 | −3.5 | 5.8 | −16.5 |
| FCHC-2-9-3 | 9738.74 | 0.935 | 21.79 | 0.34 | 0.930 | 0.892 | 0.949 | 0.929 | 0.953 | 0.6 | 4.8 | −1.5 | 0.7 | −1.8 |
| FCHC-2-9-6 | 8012.2 | 0.770 | 43.57 | 0.68 | 0.790 | 0.688 | 0.795 | 0.737 | 0.824 | −2.6 | 11.9 | −3.2 | 4.4 | −6.6 |
| FCHC-2-9-9 | 5454.75 | 0.524 | 65.36 | 1.02 | 0.589 | 0.490 | 0.585 | 0.529 | 0.648 | −11.1 | 7.0 | −10.4 | −0.9 | −19.1 |
| FCHC-2-9-12 | 3734.73 | 0.359 | 87.15 | 1.36 | 0.405 | 0.355 | 0.400 | 0.365 | 0.462 | −11.3 | 1.1 | −10.2 | −1.7 | −22.3 |
| Average | − | − | − | − | − | − | − | − | − | −0.9 | 11.8 | −0.8 | 6.8 | −7.5 |

[6] Note: For example, FCHC-2-1-3 is represented as the first group of specimens in Table 4. The material of the specimen is Q460, and the effective length is 3 m.

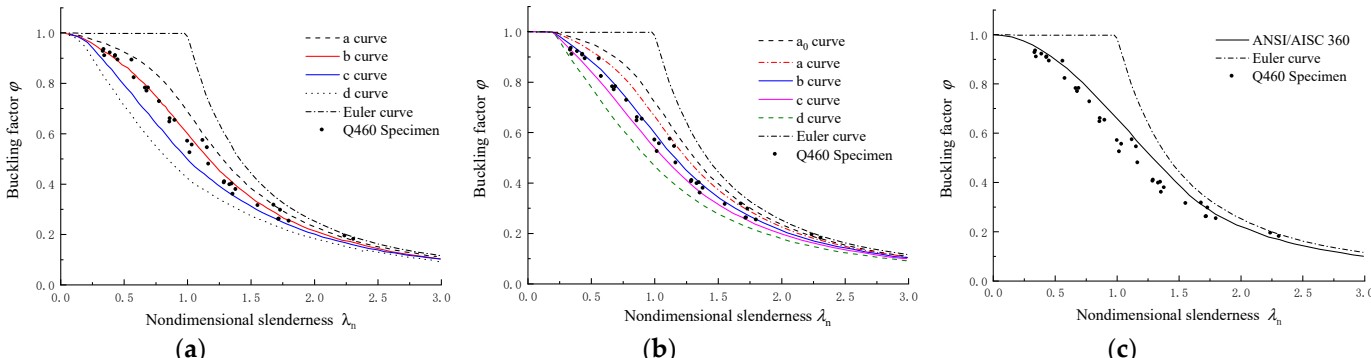

**Figure 12.** Comparison of FE results with design curves for Q460 specimens: (**a**) GB50017; (**b**) EN1993-1-1; (**c**) ANSI/AISC360.

By observing the distribution of the scattered points of the specimens in Figures 11 and 12, it is found that when the dimensionless slenderness ratio $\lambda_n$ is between 0.5 and 1.5, the scattered points are generally below the curve b; when $\lambda_n$ is less than 0.5 or greater than 1.5, the scattered points usually are above curve b. Therefore, curve b in GB 50017-2017 [16] and EN 1993-1-1 [17] are suggested for the global buckling design when $\lambda_n$ is less than 0.5 or greater than 1.5. When the $\lambda_n$ is between 0.5 and 1.5, offer curve c in GB 50017-2017 [16] and EN 1993-1-1 [17].

The reason is that curve c rather than curve b is in good agreement with ABAQUS results because the representative values of comprehensive imperfections of the class b column provided by the GB50017 standard are too large. However, the limited number of specimens is not enough to fully support the above inference, but it is conservative and safe from the designer's view.

### 5.4. Proposed Column Curves

The suggested column design curves in GB 50017-2017 [16] and EN 1993-1-1 [17] are given for the global buckling design in the previous subsection. In addition to improving the column design curves, new column curves are also proposed to predict the buckling factor of FCHCs more accurately. The expressions for the column buckling curves in both GB50017 [16] and EN 1993-1-1 [17] were derived from the Perry equation [23] and herein are applied to determine the new column design curve. Based on the expression in GB50017 [16], the fitting result for the imperfection factors were $\alpha_1 = 0.60$, $\alpha_2 = 0.970$, and $\alpha_3 = 0.320$, and the corresponding new curve (i.e., Fitting curve 1) is plotted with the results of the ABAQUS analysis in Figure 13. The fitting process was based on the average level of all ABAQUS analysis results adopted in current design codes. The result for the imperfection factor $\alpha$ based on the expression in EN 1993-1-1 [17] was 0.410, which is between that of curve b ($\alpha_b = 0.34$) and the curve c ($\alpha_c = 0.49$). The new column curve (i.e., Fitting curve 2) is also plotted in Figure 13.

Although the proposed curves above appear to agree with the ABAQUS analysis, it is still inadequate for a design code. Moreover, because any design code recommendation must be based on research, numerous practice-oriented evaluations must be examined. Consequently, more experimental and numerical investigations of FCHCs are still needed to confirm the suggested design methods in the present study, primarily analyses with more specimens made of hot-rolled H-shapes of different countries and regions.

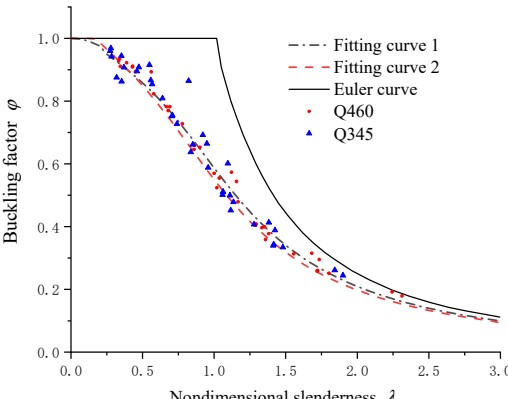

**Figure 13.** Comparison of FE results with proposed design curves.

## 6. Conclusions

The paper investigated seventy-two novel FCHCs with pin-ended supports by ABAQUS analysis. The buckling strength and deformation were determined by the verified FE Model that considers material and geometric nonlinearity and contact. Based on the ABAQUS analysis results, the conclusions are formed as follows:

1. The representative values of comprehensive imperfections given in GB50017-2017 are suitable for global buckling analysis on FCHCs. The analysis results are a little bit conservative;
2. The overall instability form of FHCs with initial imperfections is flexural buckling around the axis of symmetry, and overall torsional buckling will not occur. Hot-rolled H-shapes selected according to Chinese standards generally do not undergo local buckling;
3. When $\lambda_n \leq 0.5$ or $\lambda_n \geq 1.5$, curve b in GB 50017-2017 and EN 1993-1-1 are suggested for the global buckling design of FCHCs. When the $0.5 < \lambda_n < 1.5 \lambda_n$, take curve c in both GB 50017-2017 and EN 1993-1-1;
4. New column curves proposed in the paper to predict the buckling strength of FCHCs are accurate, but they still need more investigation to confirm. The fitting imperfection factor is $\alpha = 0.410$ based on the expression in EN 1993-1-1, and the corresponding factors based on the expression in GB50017-2017 are $\alpha_1 = 0.60$, $\alpha_2 = 0.970$, and $\alpha_3 = 0.320$.

## 7. Patents

Patent of the People's Republic of China, patent number: ZL2020 2 2612828.6 and ZL202120268874.0

**Author Contributions:** Conceptualization, L.L. and D.W.; software and calculating, Z.D. and T.L.; validation, L.L. and T.L.; resources, D.W.; writing—original draft preparation, L.L.; writing—review and editing, L.L., S.D., and H.H.; project administration, D.W.; funding acquisition, D.W. All authors have read and agreed to the published version of the manuscript.

**Funding:** This research was funded by the Central Universities Cultivation of Chang'an University (300102288201).

**Acknowledgments:** Thanks to China Construction Third Bureau Group Co., Ltd. for its financial support for the project.

**Conflicts of Interest:** The authors declare no conflict of interest.

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
