# Peer review of "Global Buckling Investigation of the Flanged Cruciform H-shapes Columns (FCHCs)"

_applsci, doi:10.3390/app112311458_

Round 1
Reviewer 1 Report
The article examines the global buckling analysis of flanged cruciform H-shape columns, which is an interesting topic that the authors have studied through several examples. The authors employed ABAQUS software to conduct finite element analyses on steel components and to obtain numerical results. Additionally, this article contrasts multiple works on the same subject. Overall this is a technically interesting study that can be reconsidered for publication at Applied Sciences if the following points are fully addressed in a revised version of the manuscript.
- The article contains a few grammatical problems; please reread it carefully possibly with the help of a mother-tongue reader. This is a major point.
- The introduction should be shortened; in its existing form, it is very long and contains some repetitive material.
- " Theoretically, the cross-sections of the three hot-rolled H-shapes that make the FCHC can be completely different." Kindly explain this statement.
- Please explain briefly how the mesh size was determined.
- Please describe the boundary conditions used in the model and provide a brief explanation in the text.
- The literature review would benefit from inclusion of the following studies, which could also be considered for a comparative analysis: Buckling behavior of curved composite beams with different elastic response in tension and compression, Composite Structures, 100, 280-289, 2013; Experimental and numerical study on the lateral-torsional buckling of steel C-beams with variable cross-section, Metals, 8(1), 941, 2018.
Author Response
1.The article contains a few grammatical problems; please reread it carefully possibly with the help of a mother-tongue reader. This is a major point.
A: Thank you, reviewer, for pointing out this problem. We have done our best to improve the grammatical problems and check by professional grammar software Grammarly (Professional Edition), and all changes are highlighted in the paper.
2.The introduction should be shortened; in its existing form, it is very long and contains some repetitive material.
A: Thank you, reviewer, for pointing out this problem.; and we have shortened the introduction, and all changes are highlighted in the paper.
3." Theoretically, the cross-sections of the three hot-rolled H-shapes that make the FCHC can be completely different." Kindly explain this statement.
A: Thank you, reviewer, we have accepted your advice, and we added the necessary explanations in the paper. All changes are highlighted in the paper.
4.Please explain briefly how the mesh size was determined.
A: Thank you, reviewer, we have accepted your advice, and we added the necessary explanations in the paper. All changes are highlighted in the paper.
5.Please describe the boundary conditions used in the model and provide a brief explanation in the text.
A: Thank you, reviewer, we have accepted your advice, and we added the necessary explanations in the paper. All changes are highlighted in the paper.
6.The literature review would benefit from inclusion of the following studies, which could also be considered for a comparative analysis: Buckling behavior of curved composite beams with different elastic response in tension and compression, Composite Structures, 100, 280-289, 2013; Experimental and numerical study on the lateral-torsional buckling of steel C-beams with variable cross-section, Metals, 8(1), 941, 2018.
A: Thank you, reviewer, we have accepted your advice, and we have cited these journal articles in the introduction and added the references [15] and [16]. All changes are highlighted in the paper.
Reviewer 2 Report
In the manuscript the authors proposed the new flanged H-shapes columns used for three hot-rolled H-shapes. The authors numerically analyzed the buckling strength and deformation of these columns under an axial compression load.
The topic of the manuscript may be of interest to scientists who deal with this field. Some excerpts from the paper may be difficult to understand by scientists not involved in this subject.
In my opinion the described results are valuable and they have application value, therefore I believe the paper is worth publishing.
Manuscript needs revisions. My specific comments and suggestions:
1) The paper needs English language editing.
2) The readers of the publication are specialists in various fields, so not all symbols in the manuscript will be understandable to them. Therefore for their needs, specialized abbreviations, acronyms and symbols should be clarified/explained the first time they are used in the text of the paper. Examples:
- the meaning of the acronym FE in line 115,
- the meaning of all symbols on the X and Y axes in Fig. 3 (σst, σu etc.).
- symbols in all tables, mathematical formulas, etc.
I suggest to complete this information in the manuscript.
Author Response
1) The paper needs English language editing.
A: Thank you, reviewer, for pointing out this problem. We have done our best to improve the grammatical problems, check by professional grammar software Grammarly (Professional Edition), and eliminate grammatical errors. Since we are not native English-speaking people, there must be some minor problems in language style, but we believe that it will not affect professional readers' understanding of article research issues. Therefore, some changes are highlighted in the paper.
2) The readers of the publication are specialists in various fields, so not all symbols in the manuscript will be understandable to them. Therefore for their needs, specialized abbreviations, acronyms and symbols should be clarified/explained the first time they are used in the text of the paper. Examples:
- the meaning of the acronym FE in line 115,
- the meaning of all symbols on the X and Y axes in Fig. 3 (σst, σu etc.).
- symbols in all tables, mathematical formulas, etc.
A: Thank you for pointing out this problem, and we have added the necessary explanations to help readers understand these symbols and acronyms. All changes are highlighted in the paper.